# Improvement in Diabetic Retinopathy through Protection against Retinal Apoptosis in Spontaneously Diabetic Torii Rats Mediated by Ethanol Extract of *Osteomeles schwerinae* C.K. Schneid

**DOI:** 10.3390/nu11030546

**Published:** 2019-03-04

**Authors:** Chan-Sik Kim, Junghyun Kim, Young Sook Kim, Kyuhyung Jo, Yun Mi Lee, Dong Ho Jung, Ik Soo Lee, Joo-Hwan Kim, Jin Sook Kim

**Affiliations:** 1Clinical Medicine Division, Korea Institute of Oriental Medicine, Daejeon 34054, Korea; chskim@kiom.re.kr (C.-S.K.); jopd7414@kiom.re.kr (K.J.); 2Korean Convergence Medicine, University of Science and Technology (UST), Daejeon 34113, Korea; ykim@kiom.re.kr; 3Herbal Medicine Research Division, Korea Institute of Oriental Medicine, Daejeon 34054, Korea; dvmhyun@jbnu.ac.kr (J.K.); candykomg@kiom.re.kr (Y.M.L.); jdh9636@kiom.re.kr (D.H.J.); knifer48@kiom.re.kr (I.S.L.); 4Department of Oral Pathology, School of Dentistry, Chonbuk National University, Jeonju 54896, Korea; 5Clinical Research Coordination Team, Korea Institute of Oriental Medicine, Daejeon 34054, Korea; 6Department of Life Science. Gachon University, 1342, Seongnamdaero, Seongnam, Gyeonggido 13120, Korea; kimjh2009@gachon.ac.kr

**Keywords:** *Osteomeles schwerinae*, diabetic retinopathy (DR), spontaneously diabetic Torii (SDT) rat, human retinal microvascular endothelial cells (HRMECs), advanced glycation end products (AGEs), retinal apoptosis, oxidative stress, mitochondrial function, adjunctive effect, combination therapy

## Abstract

Retinal apoptosis plays a critical role in the progression of diabetic retinopathy (DR), a common diabetic complication. Currently, the tight control of blood glucose levels is the standard approach to prevent or delay the progression of DR. However, prevalence of DR among diabetic patients remains high. Focusing on natural nutrients or herbal medicines that can prevent or delay the onset of diabetic complications, we administered an ethanol extract of the aerial portion of *Osteomeles schwerinae* (OSSCE), a Chinese herbal medicine, over a period of 17 weeks to spontaneously diabetic Torii (SDT) rats. OSSCE was found to ameliorate retinal apoptosis through the regulation of advanced glycation end product (AGE) accumulation, oxidative stress, and mitochondrial function via the inhibition of NF-κB activity, in turn, through the downregulation of PKCδ, P47phox, and ERK1/2. We further demonstrated in 25 mM glucose-treated human retinal microvascular endothelial cells (HRMECs) that hyperoside (3-O-galactoside-quercetin), quercitrin (3-O-rhamnoside-quercetin), and 2″-O-acetylvitexin (8-C-(2″-O-acetyl-glucoside)-apigenin) were the active components of OSSCE that mediated its pharmacological action. Our results provide evidence that OSSCE is a powerful agent that may directly mediate a delay in the development or disease improvement in patients of DR.

## 1. Introduction

Diabetic retinopathy (DR) is common long-term microvascular complication of diabetes and is a microcirculation disorder that accounts for the large majority of cases of visual impairment in working-age adults [1]. Early changes in DR include the apoptosis of peripheral blood cells, microvascular occlusion, vascular leakage, and microaneurysm [2]. Retinal endothelial cells (REC) form the first barrier that senses changes in the blood glucose. Two hallmarks of human retinal cell loss in chronic diabetes have been reported—the loss of the blood–retinal barrier integrity and direct effects on metabolism in the neural retina [3]. The diabetic metabolic influence on retinal neurons leads to an increase in apoptosis, which, in turn, causes the breakdown of the blood–retinal barrier. The retinal neuron cells begin to die soon after the onset of streptozotocin (STZ)-induced diabetes in an experimental rat model. The increase in frequency of apoptosis occurred after only one month of induction, and a similar increase was noted in human retinas after six years of diabetes [4]. The current therapy for patients with DR involves the tight control of blood glucose levels, with the aim of postponing the disease onset and progression. Nevertheless, the prevalence of DR remains high [5]. To address this problem, at least in the form of adjunct treatment, we have used natural resources as nutrients or herbal medicines to develop an alternative preventive and/or therapeutic strategy against the onset and progression of retinal apoptosis.

Under conditions of chronic hyperglycaemia, glucose and other reducing sugars react nonenzymatically with proteins, leading to the formation of advanced glycation end products (AGEs). AGEs remain tightly bound to proteins and form intra- and intermolecular crosslinks with adjacent proteins [6]. Their formation and accumulation damage cells in tissues such as the retinal vascular endothelium and kidney glomerular mesangium via the binding interactions between them and the AGE receptor (RAGE) [7,8,9,10]. In patients with diabetes, AGEs are abnormally elevated and are found to be accumulated in tissues and organs that form the sites for chronic diabetic complications [7]. In this vein, Hamme’s group reported that the AGEs accumulated in diabetic retinal vascular cells [11] promoted retinal apoptosis and vascular hyper-permeability [12]. Under hyperglycaemic conditions, oxidative stress is initiated by the generation of free radicals through protein glycation. An abnormal increase in reactive oxygen species (ROS) levels and/or a decrease in antioxidant levels leads to cellular damage by hampering normal mitochondrial function. The damaged organelles trigger the apoptotic signalling pathway [13]. Oxidative stress-induced apoptosis follows the intrinsic mitochondrial pathway, with the disruption in balance between the proapoptotic protein, B-cell lymphoma-2-associated X protein (Bax), and the antiapoptotic protein, B-cell lymphoma-1 (Bcl-1) proteins, resulting in an excess of proapoptotic proteins in the cells, which reduces the mitochondrial membrane potential (ΔΨM) following the release of cytochrome c into the cytosol [14].

*Osteomeles schwerinae* C. K. Schneid (Chinese name: Huaxixiaoshiji) is recorded in the traditional Chinese book of botanical medicine, the Chinese Materia Medica. It is a species of deciduous, semi-evergreen shrubs of the family Rosaceae that is indigenous to Asia and Polynesia. It has been used in traditional Chinese folk medicine to treat various diseases, including dysentery, diarrhoea, and so on [15]. In our preliminary studies, it was discovered that an ethanol extract of the leaves and twigs of *O. schwerinae* (OSSCE) and two flavonoids, hyperoside and quercitrin, isolated from OSSCE inhibited the activity of rat lens aldose reductase (RLAR) [16]. Specifically, a novel phytochemical compound, 5′-methoxy (1,1′-biphenyl)-3,4,3′-triol from OSSCE (referred to as K24), was confirmed to reduce the dilation of hyaloid-retinal vessels to near-normal values in 130 mM glucose-treated *flk: EGFP* (a receptor for vascular endothelial growth factor, *flk*, expressed the enhanced green fluorescent protein) transgenic zebrafish larvae [17]. The antiangiogenic action of K24 was also demonstrated in an oxygen-induced retinopathy (OIR) mouse model [18]. Another novel compound from OSSCE, 4-hydroxy-3′,5′-dimethoxybiphenyl-(1,1′-biphenyl)-3-O-β-D-glucopyranoside (referred to as K19), inhibits the nonenzymatic formation of AGE and the cross-linking of AGE to collagen in vitro. An intravitreal injection of K19 into the AGE-modified rat serum albumin (AGE-RSA)-injected Sprague-Dawley (SD) rats inhibited a retinal vascular leakage by suppressing the expression of vascular endothelial growth factor (VEGF) and by preventing the loss of occludin, an important tight junction protein [19]. We have previously reported that OSSCE reduces the AGE/RAGE binding interaction and the expression of TGF-β1 by pERK1/2, p38MAPK, and IκB phosphorylation in mouse glomerular mesangial cells under diabetic conditions [20]. Furthermore, it was also confirmed that OSSCE inhibits the extracellular matrix accumulation and mesangial proliferation of glomeruli in spontaneously diabetic Torii (SDT) rats through the inhibition of the interaction between the platelet-derived growth factor-B chain (PDGF-BB) and the PDGF-BB receptor (PDGFR-β) [21]. Hyperoside, isolated from *Abelmoschus manihot*, prevents glomerular podocyte apoptosis in STZ-induced diabetic nephropathy [22]. Hyperoside from *Allium victorialis* exhibits inhibitory effects on AGE formation and disrupts AGE-RAGE binding in hRAGE overexpressing mesangial cells [23]. 

In this study, we investigated the inhibitory effects of OSSCE on AGE accumulation and retinal cell apoptosis in SDT rats. A multi-targeted mode of action was confirmed in human retinal microvascular endothelial cells (HRMECs) for OSSCE and its marker compounds (MCs), quercitrin, hyperoside, and 2″-O-acetylvitexin under hyperglycaemic conditions.

## 2. Materials and Methods

### 2.1. OSSCE Preparation

OSSCE was collected in Kunming, Yunnan Province, China, in April 2011 and identified by Professor Joo Hwan Kim (Gachon University, Korea). A voucher specimen (no. DiAB-141) was deposited in the herbarium of the Korea Institute of Oriental Medicine (KIOM), Korea. For animal and cell studies, air-dried leaves and twigs (4 kg) were extracted with EtOH three times by maceration. The combined extracts were filtered and concentrated using a vacuum evaporator, leaving behind the EtOH extract [16].

### 2.2. High-Performance Liquid Chromatography (HPLC) Chromatogram of OSSCE

The air-dried leaves and twigs of OSSCE were chopped and then extracted with 99% ethanol for 24 h at room temperature under reflux and concentrated to obtain OSSCE. Hyperoside and quercitrin were purchased from Sigma, and 2″-O-acetylvitexin was isolated from OSSCE and was identified from the spectroscopic data. An HPLC analysis was performed using an Agilent 1200 HPLC instrument (Agilent Technologies, USA) equipped with a binary pump, vacuum degasser, auto sampler, column compartment, and diode array detector (DAD). The column used was a Luna C_18_ (250 × 4.6 mm/5.0 µm, Phenomex, USA). The mobile phase was composed of HPLC grade methanol (**A**) and 0.1% acetic acid in H_2_O (**B**) and gradually changed as follows: from 0 min to 40 min (**A**: 25%–45%; **B**: 75%–55%); from 40 min to 55 min (**A**: 45%–70%; **B** 55%–30%); from 55 min to 65 min (**A**: 70%–100%; **B** 30%–0%); and from 65 min to 70 min (**A** 100%). The column temperature was maintained at 30 °C. The analysis was performed at a flow rate of 1.0 mL/min and monitored at UV 254 nm.

### 2.3. Inhibitory Activity on Nonenzymatic AGE Formation

Bovine serum albumin (BSA; Roche Diagnostics, Basel, Swiss) in a phosphate buffer containing sodium azide (s-8032, Sigma-Aldrich, St. Louis, MO, USA) was added to a 0.2 M solution of glucose and fructose. This solution was added to the OSSCE or aminoguanidine (AG; 396494; Sigma-Aldrich), a positive control. Following 14 days of incubation, the AGE-specific fluorescence was analysed using a spectrofluorometer (Synergy HT; BIO-TEK, Winooski, VT, USA; 370 nm/440 nm). The IC_50_ (inhibitory concentration which nonenzymatic AGE formation is reduced by half) was calculated from the dose inhibition curve.

### 2.4. Inhibitory Activity on AGEs Formation Expression of RAGE in HRMECs

Human retinal microvascular endothelial cells (HRMECs) were purchased from Cell Systems (Cat. No. ACBRI 181, Kirkland, WA, USA) and used at passages 3–7. The cells were grown in a Cell Systems serum and CultureBoost^TM^ medium (CSC complete medium, CS-4ZO-500; Cell Systems) containing Bac-Off^®^ (antibiotic). The cultures were maintained at 37 °C in a humidified 95% air/5% CO_2_ atmosphere [24]. For the inhibitory activity on the AGE formation and expression of RAGE, the cells were treated with either OSSCE or AG dissolved in dimethyl sulfoxide (DMSO) for 1 h before the addition of 25 mM high glucose (HG) and 500 µg/mL BSA, following which they were incubated for 24 h. The cells were prepared for a Western blot analysis.

### 2.5. Animal Experimental Design

SDT rats 10 weeks of age and age-matched SD rats were purchased from CLEA Japan (Tokyo, Japan) and OrientBio (Korea), respectively. They were acclimated and maintained in a controlled temperature room (22 ± 2 °C in 55 ± 10% relative humidity) with a 12-h light–dark cycle. They received a basal diet (5L79, PMI Nutrition International, St Louis, MO, USA) and tap water ad libitum for 14 weeks until the blood glucose levels of the SDT rats reached 300 mg/dL. At 24 weeks of age, the rats were randomly divided into four groups: (1) normal SD rats (Nor, *n* = 10), (2) vehicle-treated SDT rats (SDT, *n* = 10), (3) SDT rats treated with 100 mg/kg/day of OSSCE (OSSCE-100, *n* = 10), and (4) SDT rats treated with 250 mg/kg/day of OSSCE (OSSCE-250; *n* = 10). The OSSCE was dissolved in distilled water and administered once a day orally for 17 weeks. All 42-week-old rats were sacrificed. All animal care procedures were approved by the Institutional Animal Care and Use Committee of KIOM. Blood samples were obtained at the time of sacrifice. The blood glucose level was measured with an automated biochemistry analyser (HITACHI 917, Japan), and the glycated haemoglobin was determined by a commercial kit (Unimate HbA1c, Roche Diagnostic, Mannheim, Germany) [21].

### 2.6. Western Blot Analysis

The cells were treated with a Laemmli sample buffer (Cat. No. 161-0737, Bio-Rad, CA, USA) and heated to 100 °C for 5 min. The proteins were electrophoresed at 20 μg/lane on a denaturing sodium dodecyl sulfate-polyacrylamide gel (SDS–PAGE) and transferred to a nitrocellulose membrane (Whatman, GmbH, Hahne str., Germany) using a Bio-Rad tank blotting apparatus (Bio-Rad, Hercules, CA, USA). The membranes were probed with 1:1000–1:2000 dilutions of primary antibodies against p47 Phox (Santa Cruz Biotechnology), ERK1/2 (Cell Signaling Technology, Danvers, MA, USA), PKC_δ_ (Santa Cruz Biotechnology), AGE (Trans Genic Inc.), RAGE (Cell signalling), and β-actin (Sigma). The membrane was washed and incubated with a horseradish peroxidase-coupled goat anti-rabbit IgG (Santa Cruz Biotechnology). After washing the membranes thrice, the signals were detected with a WEST-one enhanced chemiluminescence (ECL) solution (Intron, Korea) using a Fujifilm LAS-3000 (LAS-3000, Fuji Photo, Tokyo, Japan). The band intensities were determined using Multi Gauge Version 3.0 software.

### 2.7. Terminal Deoxynucleotidyl Transferase dUTP Nick End Labeling (TUNEL) Staining

The rat retinal vessel was fixed with 4% paraformaldehyde. The TUNEL staining was performed with a Dead End Fluorometric TUNEL kit as per the manufacturer’s instructions (Promega, Madison, WI, USA).

### 2.8. IκB Kinase (IKK) Complex Assay

The IKK activity was evaluated using an IKK-β inhibitor screening kit (Calbiochem, CA, USA) according to the manufacturer’s instructions.

### 2.9. Morphological Observation of Mitochondria

For the assessment of mitochondrial morphology in living cells, the mitochondria were stained with MitoTracker red (Life Technologies, USA) and phalloidin (Santa Cruz, USA) for 30 min at 37 °C in a humidified chamber with 5% CO_2_. Images were taken using an Olympus FV10i confocal microscope. To observe the individual mitochondria, z-stack images were acquired in series of six slices per cell ranging in thickness from 0.5 to 0.8 μm per slice.

### 2.10. Mitochondrial Membrane Potential (ΔΨm) Analysis

The lipophilic cationic probe JC-1 (Abcam, USA) was employed to measure the mitochondrial membrane potential (ΔΨm) of cells according to the manufacturer’s directions. The cells were incubated with 5 μg/mL JC-1 for 20 min and rinsed with a JC-1 staining buffer. The fluorescence intensity of mitochondrial JC-1 monomers (green) and aggregates (red) was detected using an Olympus microscope (BX51, Olympus, Japan). In healthy cells with high mitochondrial ΔΨm, JC-1 forms complexes that emit intense red fluorescence (JC-1 aggregates). In apoptotic cells with low ΔΨm, JC-1 remains in the monomeric form and emits a green fluorescence. The ratio of red to green fluorescence was calculated by analysing the digital images using Image J software (National Institutes of Health, MD, USA) and was indicative of the ΔΨm.

### 2.11. Intracellular ROS Measurement

The measurement of intracellular ROS levels was made using dihydrodichlorofluorescein diacetate (DCF-DA) in which the fluorescent probe, 2′,7′-dichlorodihydrofluorescein diacetate (H_2_DCF-DA; Molecular Probes Inc., Eugene, OR, USA), was converted by intracellular esterase to H_2_DCF, which was oxidized by intracellular ROS to the highly fluorescent DCF. The OSSCE or MC treatment was administered for 10 min, and the cells were then stimulated with HG for 96 h. The cells were washed with Hank’s Balanced Salt Solution (HBSS) buffer and incubated in the dark for 30 min in HBSS buffer containing 50 µM H_2_DCF-DA. The DCF fluorescence was measured using a Synergy HT spectrofluorometer (excitation 485 nm/emission 530 nm, BIO-TEK, VT, USA). The production of intracellular ROS was visualized by the fluorescence microscopic imaging of cells incubated in the dark for 5 min in a HBSS buffer containing 10 µM H_2_DCF-DA, using an Olympus microscope (BX51, Olympus, Japan) equipped with an Olympus DP 70 camera.

### 2.12. Intracellular 8-OHdG Measurements

The cells were washed with PBS, fixed, and permeabilized with 0.2% Triton X-100. Following three additional washes, the cells were incubated with a primary antibody against 8-OHdG (1:100, Abcam), washed, and incubated with the secondary antibody conjugated to Alexa Fluor 594. After removing the secondary antibody, the cells were washed three times and observed under the inverted fluorescence microscope.

### 2.13. Immunostaining

The cells were grown to 80% confluency in 4-well slides, synchronized, and exposed for 96 h to HG in the absence or presence of the treatment solution (OSSCE or MCs). The cells were fixed for 15 min in 4% paraformaldehyde in PBS at 4 °C and washed. For the determination of NF-kB nuclear translocation, the treated HRMECs were washed and fixed using 4% paraformaldehyde in PBS. The cells were then washed and treated with 10% goat serum in PBS for 30 min to block nonspecific binding. The primary NF-kB p65 antibody (1:200, #8242, Cell signalling) was diluted 1:1000 and incubated for 1 h. After further washing, the cells were incubated with fluorescein isothiocyanate (FITC) labeled antibody for 1 h. The stained cells were sealed with a mounting solution (DAKO, Glostrup, Denmark) and observed using an Olympus fluorescence microscope (BX51) equipped with an Olympus DP 70 camera. Immunohistochemistry was performed as previously described [25]. The following antibodies were used: Monoclonal mouse anti-AGEs (1:200, cat. no. KAL-KH001; Cosmo Bio Co, Ltd., Tokyo, Japan). For the detection of the AGEs, the sections were incubated with a labeled streptavidin-biotin kit (DAKO, Carpinteria, CA, USA) and were visualized by 3,3′-diaminobenzidine tetrahydrochloride. The images were captured using an Olympus BX51 microscope and DP71 digital camera (Olympus). For the morphometric analysis, the positive signal intensity per unit area (0.32mm^2^) in a total of 5 randomly selected fields were determined using Image J software (version 1.52; National Institutes of Health, Bethesda, MD, USA).

### 2.14. Measuring Nuclear Factor-κB (NF-κB) Activity

For the electrophoretic mobility shift assay (EMSA), nuclear extracts were prepared with a kit according to the manufacturer’s instructions (NE-PER™ nuclear and cytoplasmic extraction reagents; Pierce Biotechnology, Inc., Rockford, IL, USA). The EMSA assay was performed by incubating 10 μg nuclear protein extract with IRDye 700-labeled NF-κB oligonucleotide (LI-COR Biosciences, Lincoln, NE, USA) or an unlabeled NF-κB probe (Promega Corporation) for cold competition. The EMSA gels were analyzed, and the images were captured and quantified using a LI-COR Odyssey infrared laser imaging system (LI-COR Biosciences).

### 2.15. Measurement of NADPH Oxidase Activity

After treatment with OSSCE or MCs, the cells were washed, scraped, and then harvested with a lysis buffer containing 20 mM KH_2_PO_4_, protease mixture inhibitor, 1 mM EGTA, 10 μg/mL aprotinin, and 0.5 μg/mL phenylmethane sulfonyl fluoride (PMSF) at 4 °C. Following centrifugation at 10,000 *g* for 10 min, the cell lysates were analysed immediately [26].

### 2.16. Statistical Analysis

Image analysis was implemented using Image J software (National Institutes of Health, MD, USA) and averaged. All experiments were repeated at least three times. The data are analysed using a one-way analysis of variance (ANOVA) followed by Tukey’s multiple comparison test or using an unpaired Student’s *t*-test with the Prism 6.0 software (GraphPad software, San Diego, CA, USA).

## 3. Results

### 3.1. HPLC Chromatogram of OSSCE

The HPLC analysis demonstrated that hyperoside, quercitrin, and 2″-O-acetylvitexin are marker compounds for OSSCE (Figure 1).

### 3.2. OSSCE Inhibits Nonenzymatic AGE Formation In Vitro, Expressions of Ages and RAGE in 25 mm Glucose-Treated HRMECs, and AGE Level in Serum and Whole Retina of SDT Rats

OSSCE inhibits the nonenzymatic formation of AGEs (IC_50_: 16.34 ± 0.04 µg/mL) more effectively than aminoguanidine (AG), an established AGE inhibitor (IC_50_: 72.28 ± 4.21 µg/mL) (Figure 2a). HRMECs were treated with 10 ng/mL OSSCE or 10 nM doses of the three identified MCs and then incubated with 25 mM glucose (HG). OSSCE- and MC-treated HRMECs showed a marked reduction in the formation of AGEs compared with vehicle-treated HRMECs (^###^
*p* < 0.001). OSSCE and quercitrin significantly reduced the expression of RAGE (^##^
*p* < 0.01, ^#^
*p* < 0.05). The RAGE expression in hyperoside- and 2″-O-acetylvitexin-treated groups exhibited a decreasing trend (Figure 2b). The concentration of serum AGEs was prominently increased in vehicle-treated SDT rats compared with normal SD rats (** *p* < 0.01). The OSSCE treatment (250 mg/kg/day) significantly decreased AGE levels in SDT rats relative to vehicle-treated rats (^#^
*p* < 0.05) (Figure 2c). Whole retinal tissue from vehicle-treated SDT rats showed a significant accumulation of AGEs relative to normal SD rats (** *p* < 0.01). High doses of OSSCE significantly reduced the levels of AGEs relative to the levels in vehicle-treated SDT rats (^#^
*p* < 0.05) (Figure 2d).

### 3.3. OSSCE Inhibits Apoptosis of the Retinal Ganglion Cell Layer and Whole Retinal Vessels in SDT Rats

To confirm the inhibitory effect of OSSCE on retinal damage, we investigated the levels of apoptosis in SDT rat tissues. We applied the terminal deoxynucleotidyl transferase dUTP nick end labelling (TUNEL) assay in trypsin-digested retinal ganglion cells and in whole retinal vessels. The retinal trypsin digests were analysed to quantitate TUNEL-positive cells. The examination of the retinal trypsin digests of vehicle-treated SDT rats showed dramatic increases in TUNEL-positive cells in the retinal ganglia (** *p* < 0.01) and in whole retinal vessels (*** *p* < 0.001) relative to that seen in normal SD rats. The OSSCE-treated SDT rats exhibited a significant reduction in the number of TUNEL-positive cells in the ganglion layer (^#^
*p* < 0.05) relative to vehicle-treated SDT rats (Figure 3a). The levels of apoptosis in whole retinal vessels of SDT rats treated with two different dosages of OSSCE (18 ± 11%, 11 ± 9%) reduced by 67% and 78% respectively relative to the levels seen in vehicle-treated SDT rats (45 ± 15%) (^##^
*p* < 0.01; ^###^
*p* < 0.001) (Figure 3b). We investigated further the ratio between Bax and Bcl-2 and the expression of cleaved caspase-3 in the trypsin-digested whole retina of SDT rats. The ratio of Bax to Bcl-2 in vehicle-treated SDT rats was significantly increased relative to that seen in normal SD rats (** *p* < 0.01). The OSSCE-treated SDT rats exhibited a significantly reduced Bax to Bcl-2 ratio when compared with vehicle-treated SDT rats, with the decrease occurring in a dose-dependent manner (^#^
*p* < 0.05; ^##^
*p* < 0.01) (Figure 3c, left panel). The expression of cleaved caspase-3 in vehicle-treated SDT rats also increased markedly (** *p* < 0.01) but was significantly decreased in 250 mg/kg OSSCE-treated SDT rats (^#^
*p* < 0.05) (Figure 3c, right panel).

### 3.4. OSSCE and MCs Inhibit HG-induced Intracellular ROS Generation and 8-OHdG Expression in HRMECs

The HRMECs were treated with 10 ng/mL OSSCE and 10 nM MCs before incubation with HG for 96 h and assayed for intracellular ROS generation and 8-OHdG expression using a fluorescence microscopy. The HG-treated group demonstrated a significant increase in ROS generation compared with the normal group (*** *p* < 0.001). The OSSCE- and MCs-treated groups exhibited significantly lower ROS production relative to the HG-treated group (^###^
*p* < 0.001) (Figure 3d). The expression of 8-OHdG by HG was also significantly increased almost ten-fold compared to that seen in the control group (*** *p* < 0.001). The OSSCE- and MCs-treated groups exhibited significantly reduced expression of 8-OHdG when compared with the HG-treated group (^###^
*p* < 0.001) (Figure 3e).

### 3.5. Protective Effects of OSSCE and MCs on HG-Induced Mitochondrial Morphology and Mitochondrial Membrane Potential (ΔΨM) in HRMECs

The mitochondrial tubules in the HG-treated group became shorter and more fragmented compared to those from the untreated group. However, OSSCE and the MCs were found to prevent such mitochondrial damage (Figure 4a). We evaluated the effect of OSSCE and MCs on HG-induced ΔΨM in HRMECs by detecting different fluorescences emitted by monomeric and aggregated 5, 5′, 6, 6′-tetrachloro-1, 1′, 3, 3′-tetramethyl benzimidazolyl carbocyanine iodide (JC-1). The depolarization of the mitochondrial membrane was evidenced by the green fluorescence emitted by HG-treated cells, resulting from the presence of JC-1 in monomeric form. Untreated cells, on the other hand, emitted a red fluorescence due to the aggregation of JC-1. The reduced red/green fluorescence intensity ratio can, thus, indicate the depolarization of mitochondria. Figure 4b (left panel) indicates that HREMCs exposed to HG for 24 h exhibited a significant decrease in the aggregate form (red fluorescence) and an increase in the monomeric form (green fluorescence) of JC-1. However, treatment with OSSCEs and MCs prevented the loss of aggregation and the concurrent increase in monomers. In addition, as seen in the right panel of Figure 4b, the HG-treated group showed a greater variation in ΔΨM, as inferred from the lower range of red (hyperpolarized) to green (depolarized) colors when compared with that seen in normal cells (*p**** < 0.001). However, treatment with OSSCE and MCs was found to result in a 43.2%, 39.3%, 36.1%, and 48.9% increase in the red/green fluorescence ratio of JC-1 respectively, relative to that seen in HG-treated HRMECs (^##^
*p* < 0.01; ^#^
*p* < 0.05; ^#^
*p* < 0.05; ^###^
*p* < 0.001).

### 3.6. Effects of OSSCE and MCs on Mitochondria-Dependent Apoptotic Pathways in HG-Treated Hrmecs

The upregulation of mitochondrial Bax (Figure 5a, left panel), cytosolic cytochrome c (Figure 5b, right panel), and cleaved caspase-9 and -3 (Figure 5c) and the downregulation of cytosolic Bax (Figure 5a, lower, right) and mitochondrial cytochrome c (Figure 5b, left panel) were observed in HG-treated HRMECs (** *p* < 0.01; *** *p* < 0.001). OSSCE- and MCs-treated HRMECs showed a significant reversal of these effects on protein expression (^#^
*p* < 0.05; ^##^
*p* < 0.01; ^###^
*p* < 0.001).

### 3.7. OSSCE Inhibits the Activation of Nuclear Factor Κ-B (NF-Κb) in SDT Rat Retina and HG-Treated Hrmecs 

An electrophoretic mobility shift assay (EMSA) analysis of nuclear proteins revealed that OSSCE treatment at a dose of 250 mg/kg/day significantly reduced nuclear translocation and DNA-binding activity of NF-κB (^###^
*p* < 0.001), whereas vehicle-treated SDT rats resulted in increased NF-κB translocation (*** *p* < 0.001) relative to levels in normal SD rats (Figure 6a).

OSSCE and MCs treatment also prevented the nuclear translocation of NF-κB in HG-treated HRMECs (Figure 6b). The visualization (Figure 6b, left panel) and qualitative analysis of the nuclear translocation (Figure 6b, right panel) of NF-κB in HG-treated HRMECs was performed using a fluorescence microscopy and the Image J software respectively (*** *p* < 0.001). The nuclear NF-kB levels in OSSCE and MCs-treated groups of HRMECs were significantly lower than that of the HG-treated group (^###^
*p* < 0.001). We checked whether OSSCE inhibited IKK activity. As shown in Figure 6c, OSSCE treatment at doses of 50 ng/mL and 100 ng/mL dose-dependently inhibited IKK activity in HRMECs (** *p* < 0.01, *** *p* < 0.001).

### 3.8. Effects of OSSCE and MCs on NADPH Oxidase Activity and the Related Signalling Pathways in HG-Treated HRMECs

In HG-treated cells, protein kinase C (PKC) δ was dramatically activated, although PKCα/βII and PKCζ/λ were not phosphorylated (Figure 7a). HG-induced NADPH oxidase activity was significantly decreased by diphenyleneiodonium (DPI; NADPH oxidase inhibitor), rottlerin (PKCδ inhibitor), and GFX (PKC inhibitor), whereas Gö 6983 (PKCα/βII inhibitor) caused no such effect (Figure 7b). We checked whether OSSCE and its MCs can regulate the activity of NADPH oxidase. NADPH oxidase in the HG-treated group was activated (*** *p* < 0.001) when compared with the control group. OSSCE and MCs-treated groups showed a significantly reduced activity of NADPH oxidase compared with that seen in the HG-treated group (^#^
*p* < 0.05, ^##^
*p* < 0.01) (Figure 7c).

Next, we examined the inhibitory effects of OSSCE and MCs on HG-induced p47^phox^, extracellular regulated kinase (ERK)-1/2, and PKCδ expression. The HG-treated group showed a significantly elevated expression of PKCδ and p47^phox^ compared to the control group (** *p* < 0.01, *** *p* < 0.001). Treatment with OSSCE and MCs significantly downregulated PKCδ and p47^phox^ (^###^
*p* < 0.001). The upregulated ERK1/2 expression caused by HG treatment was reversed by subsequent treatment with OSSCE and MCs (^##^
*p* < 0.01, ^#^
*p* < 0.05) (Figure 7d).

### 3.9. Levels of Haemoglobin A1c (Hba1c) and Blood Glucose in SDT Rats

As already reported [21], the levels of HbA1c and blood glucose were significantly elevated in vehicle-treated SDT rats. However, these parameters in the OSSCE-treated group showed the tendency to be decreased (Table 1).

## 4. Discussion

DR is a frequent diabetic microvascular complication and one of the most common causes of legal blindness in the world. The low success of current therapeutic strategies in combating this problem points to an unmet clinical need for therapy that may slow or halt the progression of DR. It is well-known that in clinical practice, the development of diabetic complications is seen in a large number of patients even after the strict control of blood glucose by oral medications, insulin therapy [27], or use of the insulin pump [28]. Clearly, there is an urgent need for the development of alternative therapeutic approaches. Matsuda’s group suggested a pancreatic transplantation before the “point of no return”, thereby preventing or curing diabetic complications [29]. Traditional herbal medicine, sometimes as adjunctive therapy, has been demonstrated to accrue various benefits to patients suffering from a range diseases and complications [30,31]. The aim of the present study is to develop a drug candidate from herbal medicine or plant resources as a therapeutic or adjunctive approach to prevent or delay the onset of DR, including in our consideration, substances that may act through a mechanism other than the tight modulation of blood glucose levels. We investigated the potential of OSSCE against retinal apoptosis in SDT rats over a period of 17 weeks. Further, the multi-targeted mode of actions for OSSCE and its MCs—hyperoside, quercitrin, and 2″-O-acetylvitexin—were also investigated in HG-treated HRMECs.

The SDT rat spontaneously develops hyperglycaemia as a result of reduced insulin secretion due to the dysfunction of pancreatic islet tissues [32,33]. It has been frequently used as a suitable animal model for DR. Retinal vascular leakage, vascular cell loss, and proliferative neovascularization are characteristics of SDT rats that resemble the clinical features of human DR [27,33]. Matsuda’s group has reported that the non-perfusion area and neovascularization in the retina were detected at 5 weeks following the onset of diabetes in SDT rats. A leakage of the retinal vessels was also observed at 10 weeks post-onset of diabetes in SDT rats. Daily insulin treatment could not prevent or reverse these ocular changes. With regards to pancreatic transplantation, DR and diabetic cataract cannot be prevented or improved by performing a pancreatic transplantation at or beyond 10 weeks post-onset [29]. 

Hyperglycaemia leads to the formation and accumulation of irreversible AGEs, which is already known to be one of the risk factors for the progression of diabetic complications such as DR. AGEs also induce apoptotic cell death of pericytes through binding interactions with RAGE [34,35,36]. The IC_50_ value of OSSCE against nonenzymatic AGE formation (16.31 ± 0.04 µg/mL) was superior to that of aminoguanidine (AG; 72.28 ± 4.21 µg/mL), a well-known AGE inhibitor [37]. Further, it was confirmed that OSSCE and MCs significantly suppressed AGE formation and RAGE expression in HG-treated HRMECs (Figure 2b). Moreover, OSSCE-treated SDT rats showed a significant reduction in AGEs levels in the serum and whole retina (Figure 2c,d). AGE quantitative measurements following OSSCE treatment under HG conditions yielded coinciding results in both in vitro and in vivo contexts. 

Hyperglycaemia induces the activation of protein kinase C (PKC) and NADPH-oxidase, which leads to the production of ROS and oxidative stress in diabetic patients. PKC and NADPH-oxidase have been suggested as potential therapeutic targets for the control of hyperglycaemia-induced oxidative stress [38]. An increased ROS production and cellular death are related. Their association is mediated by a pathological cell death pathway (apoptosis) and may be aggravated by the interaction of AGEs with RAGEs [38]. Therefore, we evaluated the effect of OSSCE on apoptosis in SDT rat retina and the associated molecular mechanisms in HG-treated HRMECs. OSSCE also exhibited an antiapoptotic effect in the retinal ganglion cell layer (Figure 3a arrow) and whole retinal vessels of SDT rats (Figure 3b). We further investigated whether OSSCE could regulate apoptotic proteins in the SDT rat retina. Bax/Bcl-2 ratio and the level of caspase-3 were increased more than two-fold in vehicle-treated SDT rat retinas when compared to normal SD rat retinas. These abnormal increases were significantly reversed by OSSCE treatment. Particularly, at a dosage of 250 mg/kg, they were reduced to nearly normal values (Figure 3c). Intracellular ROS generation and an increased expression of 8-OHdG in HG-treated HRMECs were prevented by the administration of OSSCE (Figure 3c,d). MCs were shown to be active against oxidative stress. OSSCE reduced hyperglycaemia-induced oxidative stress, thus preventing retinal apoptosis. Oxidative stress results in the alteration of mitochondrial shape and function. The change in mitochondrial shape has been linked to neurodegeneration, reduced lifespan, and cell death [39]. The dissipation of mitochondrial integrity is one of the early events leading to apoptosis [40]. Mitochondrial dysfunction is a common denominator in several chronic nervous system diseases and diabetes [41], as well as in ischemic brain injury [42].

Hyperglycaemia-induced oxidative stress increases Bax/Bcl-2 ratio, augmenting the release of cytochrome c from mitochondria to cytosol and inducing the formation of the apoptosome. Further, it leads to the conversion of inactive procaspase 9 into active caspase 9 and procaspase 3 into caspase 3 [43]. OSSCE treatment ameliorated the damage to mitochondrial morphology and ΔΨM caused by HG in HRMECs. MCs were shown to be the active components of OSSCE responsible for this effect (Figure 4a,b). OSSCE and MCs were effective in preventing the activation of the mitochondrial-dependent apoptotic pathway in HG-treated HRMECs. HG-triggered apoptosis in HRMECs occurs via the activation of caspase-9 and -3, the enhancement of cytochrome C release into cytosol, and the subsequent interruption of the Bax/Bcl-2 balance. These detrimental effects were prevented by OSSCE and MCs (Figure 5a–c). The oxidative stress-mediated activation of NF-κB leads to the translocation of its p65 submit to the nucleus by releasing it from the inhibitory protein Iκ-Bα through Iκ-B phosphorylation [14]. The nuclear translocation of NF-kB in SDT rat retinas was significantly decreased by OSSCE treatment (250 mg/kg) (Figure 6a). In HG-treated HRMECs, OSSCE and MCs showed marked inhibition of NF-κB translocation into the nucleus (Figure 6b). HG-induced IκB kinase (IKK) activity was also dose-dependently decreased by OSSCE (50 ng/mL, 100 ng/mL) (Figure 6c). That is, OSSCE was able to inhibit NF-kB translocation through the suppression of phosphorylation and through the degradation of IκB. OSSCE, thus, acts as an IKK inhibitor. NADPH oxidase is an enzyme that catalyses the production of superoxide (O_2_^−^) from oxygen and NADPH. Superoxide produced by NADPH oxidase plays a critical role in diverse vascular diseases such as diabetic microvascular complications [44], stroke [45,46], and cardiovascular disease [47,48]. The activation of PKCδ and NADPH-oxidase ultimately leads to oxidative stress- and NF-kB-mediated apoptosis [38]. In the present study, among the PKC isoforms, only PKCδ was dramatically activated by HG in HRMECs (Figure 7a). The increase in NADPH oxidase activity mediated by HG was significantly decreased by DPI, Rottlerin, and GFX but not by Gö 6983 (Figure 7b). These data demonstrated that PKCδ plays a crucial role in the activity of NADPH oxidase in HG-treated HRMECs. OSSCE and MCs significantly inhibited the NADPH-oxidase activity by mediating a reduction in the PKC_δ_ activity (Figure 7c). The increased expression of PKCδ, the p47^phox^ subunit of NADPH-oxidase, and ERK1/2 in HG-treated HRMECs was significantly reversed by treatment with OSSCE and MCs (Figure 7d). 

## 5. Conclusions

This series of experiments strongly indicated that OSSCE mediates the protection against retinal apoptosis resulting from hyperglycaemia by simultaneously modulating AGE levels, oxidative stress-induced retinal apoptosis, and mitochondrial dysfunction through the inhibition of NF-κB translocation into the nucleus via the downregulation of PKCδ, P47^phox^ subunit of NADPH oxidase, and ERK1/2, although OSSCE, itself, could not properly control the levels of blood glucose and HbA1c in SDT rats. Taken together, we can postulate that a delay and/or prevention of the development of DR might be possible if the combination of additional OSSCE and an anti-glycaemic drug such as metformin is given to patients with diabetes before the point of no return.

## 6. Patents

The patents related to this study were registered in Korea (no. 10-097394), Hong Kong (no. HK1170958), England, France, Swiss, Germany (no. 247483), China (no. ZL 200980160639.3), the United Arab Emirates (no. 1028), and the USA (no. 8,784,911).

## Figures and Tables

**Figure 1 nutrients-11-00546-f001:**
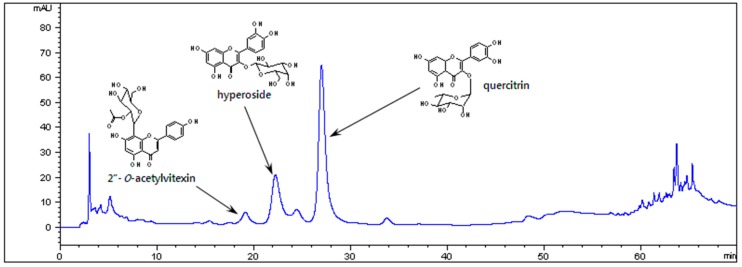
High-Performance Liquid Chromatography (HPLC) chromatogram of ethanol extract of the aerial part of *Osteomeles schwerinae* (OSSCE).

**Figure 2 nutrients-11-00546-f002:**
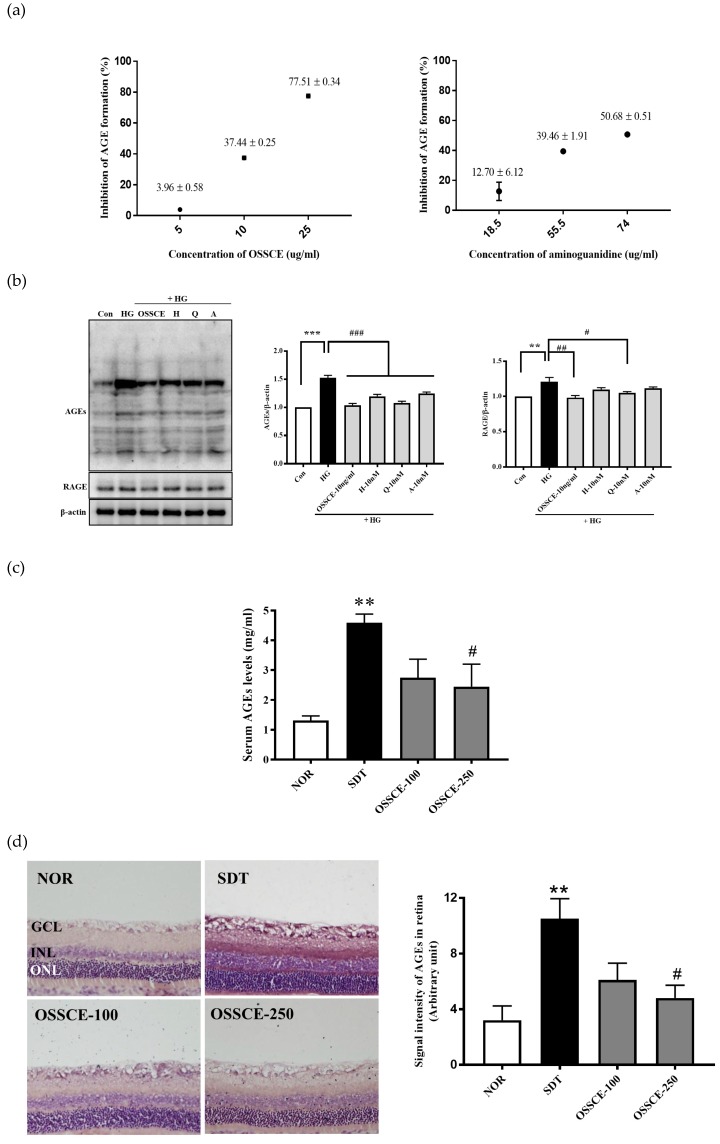
OSSCE inhibits AGE formation and RAGE expression. (**a**) The inhibitory action of OSSCE on nonenzymatic AGE formation: Aminoguanidine (AG) was used as a positive control. OSSCE was added into the solution of bovine serum albumin (BSA) and 0.2 M glucose and fructose and incubated for 14 days; the AGE-specific fluorescence was analysed using a spectrofluorometer. The IC_50_ value was calculated from the dose inhibition curve. The IC_50_ values of OSSCE and AG activity against nonenzymatic AGE formation are 16.34 ± 0.04 µg/mL and 72.28 ± 4.21 µg/mL, respectively (*n* = 3). (**b**) The inhibitory effect of OSSCE and marker compounds (MCs) on AGE formation and RAGE expression in HG-treated human retinal microvascular endothelial cells (HRMECs). Con, HG, H, Q, and A stand for control, 25 mM glucose, hyperoside, quercitrin, and 2″-O-acetylvitexin, respectively. HG incubation for 96 h was performed after treatment with OSSCE or MCs. The cell lysate was subjected to western blotting with monoclonal antibodies against specific AGEs, RAGE, and β actin, as described in the Materials Section. All data are expressed as the mean ± SD (*n* = 3). *** *p* < 0.001, ** *p* < 0.01 vs. Con.; *^###^ p* < 0.001, *^##^ p* < 0.01, *^#^ p* < 0.05 vs. HG. The AGE level in serum (**c**) and whole retina (**d**) of SDT rats: the OSSCE was administered at 100 or 250 mg/kg/day orally for 17 weeks. The serum AGE levels were analysed by enzyme-linked immunosorbent assay (ELISA). The AGEs in rat retinas were analysed by immunohistochemistry followed by densitometric quantification. GCL, ganglion cell layer; INL, inner nuclear layer; ONL, outer nuclear layer ** *p* < 0.01 vs. NOR; *^#^ p* < 0.05 vs. SDT (*n* = 3–5). The data are expressed as means ± S.D.

**Figure 3 nutrients-11-00546-f003:**
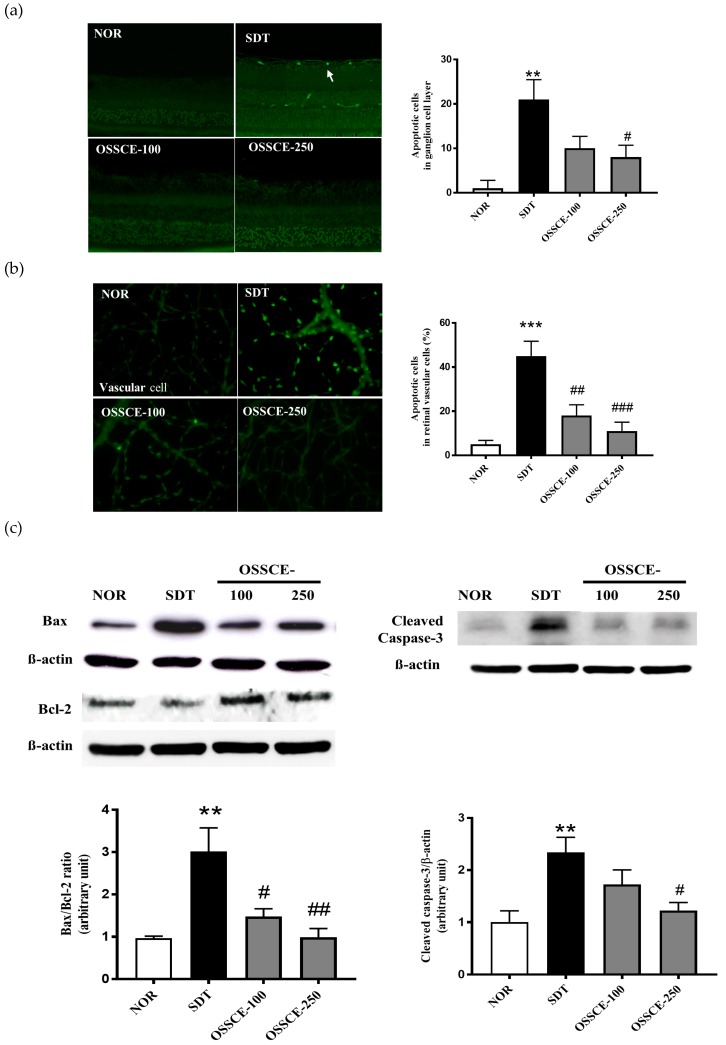
OSSCE inhibits retinal apoptosis in SDT rats, as well as ROS generation and 8-OHdG expression in HG-treated HRMECs: (**a**) The OSSCE-treated groups reduce apoptosis in retinal ganglion cells and in (**b**) retinal microvascular vessels. The retinal sections and whole mount of retinal microvascular cells from all groups were stained with a terminal deoxynucleotidyl transferase dUTP nick end labelling (TUNEL) kit; then, the TUNEL-positive cells were analysed. *** *p* < 0.001, ** *p* < 0.01 vs. NOR; ^#^
*p* < 0.05, ^##^
*p* < 0.01, ^###^
*p* < 0.001 vs. SDT. (**c**) The retinal sections in all groups were stained with Bax, Bcl-2, and cleaved caspase-3 antibodies, and their expression levels were measured quantitatively by a western blot. The ratio of Bax to Bcl-2 and the levels of cleaved caspase-3 in SDT rat retinas increased in vehicle-treated SDT rats but were significantly suppressed by OSSCE. ** *p* < 0.01 vs. NOR; ^#^
*p* < 0.05, ^##^
*p* < 0.01 vs. SDT (*n* = 3–5). (**d**) The intracellular ROS levels were measured by the dihydrodichlorofluorescein diacetate (DCF-DA). The increase observed in HG-treated HRMECs was significantly reversed by treated with OSSCE or MCs. *** *p* < 0.001 vs. Con; ^###^
*p* < 0.001 vs. HG. The data are expressed as means ± S.D. (*n* = 3). (**e**) The cells were incubated with an 8-OHdG-specific primary antibody and an Alexa Fluor 594 anti-rabbit antibody. The 8-OHdG expression was significantly decreased by treatment with OSSCE or MCs in HG-treated HRMECs. *** *p* < 0.001 vs. Con; ^###^
*p* < 0.001 vs. HG. The data are expressed as means ± S.D. (*n* = 3).

**Figure 4 nutrients-11-00546-f004:**
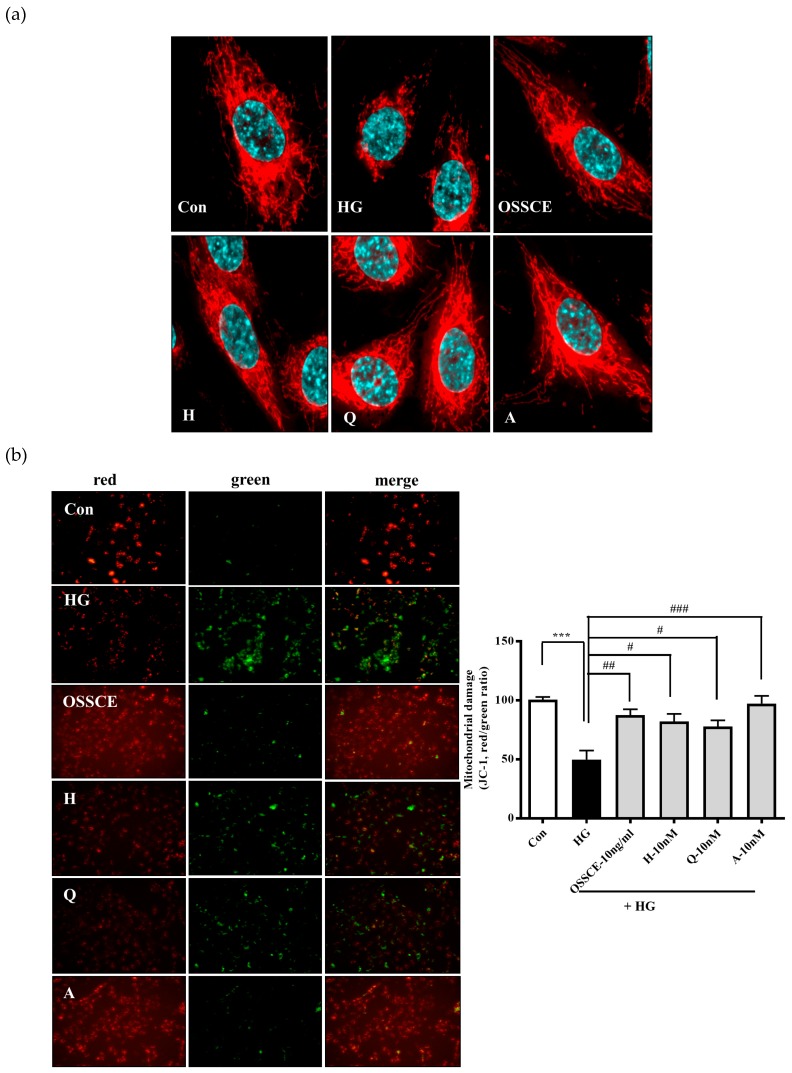
OSSCE and MCs inhibit the alteration of HG-induced mitochondrial morphology and mitochondrial membrane potential (MMP) activity in HRMECs. (**a**) OSSCE and MCs inhibit the alteration of mitochondrial shape in HG-induced HRMECs. (**b, left panel**) HRMECs were pre-incubated with OSSCE or MCs for 24 h in the absence or presence of HG, and then, the MMP was evaluated using JC-1. (**b, right panel**) The MMP was determined using an automatic fluorescence microplate reader. The MMP (ratio of red/green) activity in the OSSCE- and MC-treated groups showed significant increases compared with that in the HG-treated group, respectively. The red/green ratio (ΔΨm) of HG, OSSCE, H, Q, and A was 49.75 ± 15.49, 87.61 ± 9.59, 82.01 ± 13.22, 77.87 ± 10.38, and 97.33 ± 12.96, respectively. *** *p* < 0.001 vs. Con; ^#^
*p* < 0.05, ^##^
*p* < 0.01, ^###^
*p* < 0.001 vs. HG. The data are expressed as mean ± S.D. (*n* = 3).

**Figure 5 nutrients-11-00546-f005:**
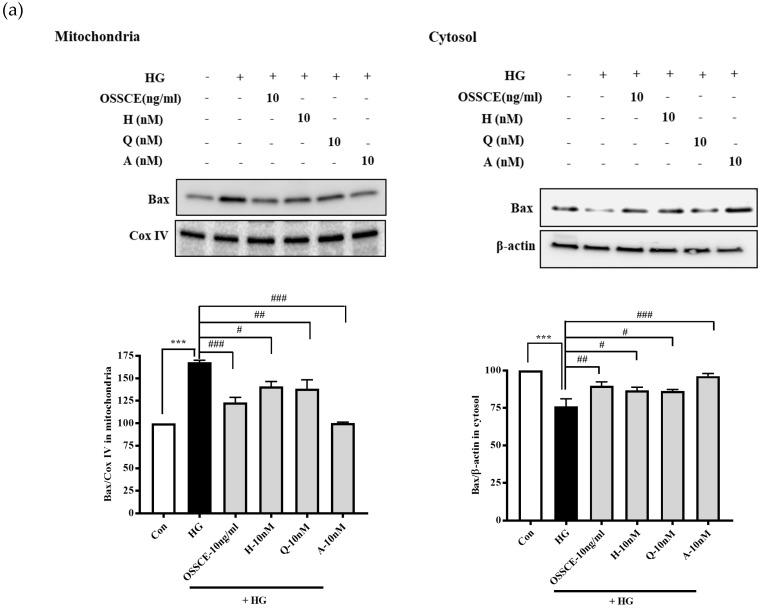
The effects of OSSCE and MCs on the mitochondria dependent-apoptotic pathways in HG-treated HRMECs: OSSCE and MCs restore the expression of Bax (**a**), cytochrome C (**b**), and caspase-9 and -3 (**c**) abnormally changed in HG-treated HRMECs. ** *p* < 0.01, *** *p* < 0.001 vs. Con; ^#^
*p* < 0.05, ^##^
*p* < 0.01, ^###^
*p* < 0.001 vs. HG. The data are expressed as mean ± SD (*n* = 3).

**Figure 6 nutrients-11-00546-f006:**
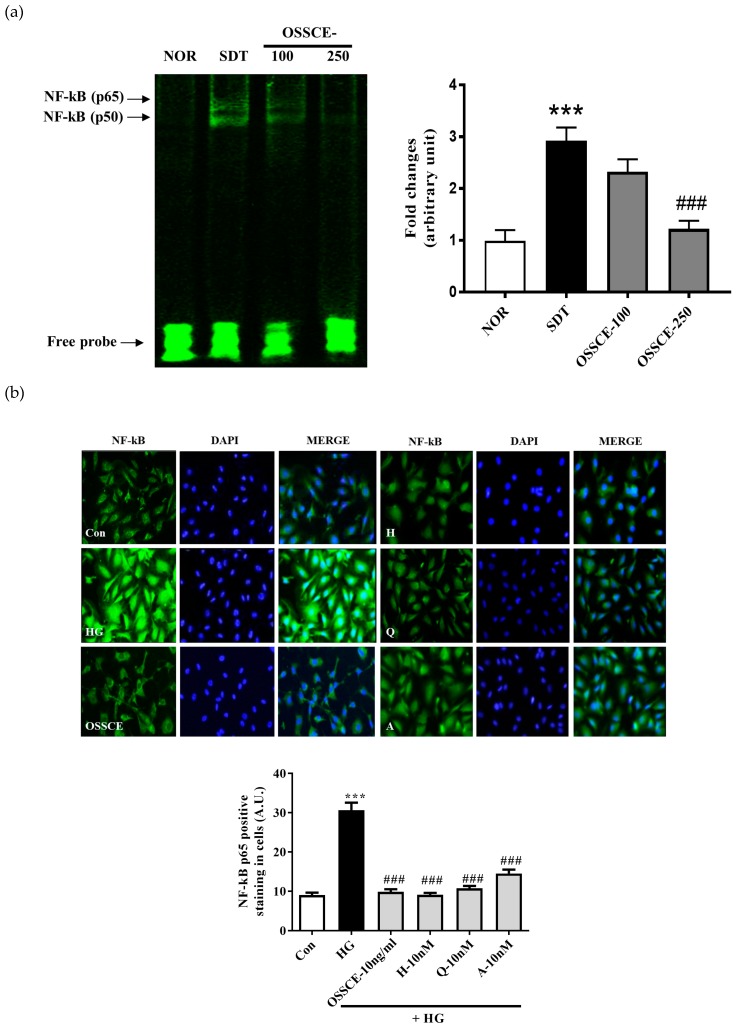
OSSCE inhibits the activation of NF-κB in the retina of SDT rats and in HG-treated HRMECs and the activity of IKK-kinase. (**a**) The increased NF-κB activity in the retina of SDT rats was decreased by OSSCE. The NF-κB activity was measured by TUNEL staining. *** *p* < 0.001 vs. NOR; ^###^
*p* < 0.001 vs. SDT (n = 3–5). (**b**) OSSCE and MCs suppressed NF-kB translocation into the nucleus in HG-treated HRMECs. *** *p* < 0.001 vs. Con; ^###^
*p* < 0.001 vs. HG (n = 4). (**c**) The inhibitory effect of OSSCE on the IκB kinase activity: The treatment with OSSCE at concentrations of 50 and 100 ng/mL inhibited the IκB kinase activity dose-dependently. IKK-2 inhibitor IV (20 ng/mL) inhibited IKK activity. ** *p* < 0.01, *** *p* < 0.001 vs. HG. The data are expressed as mean ± S.D. (*n* = 3).

**Figure 7 nutrients-11-00546-f007:**
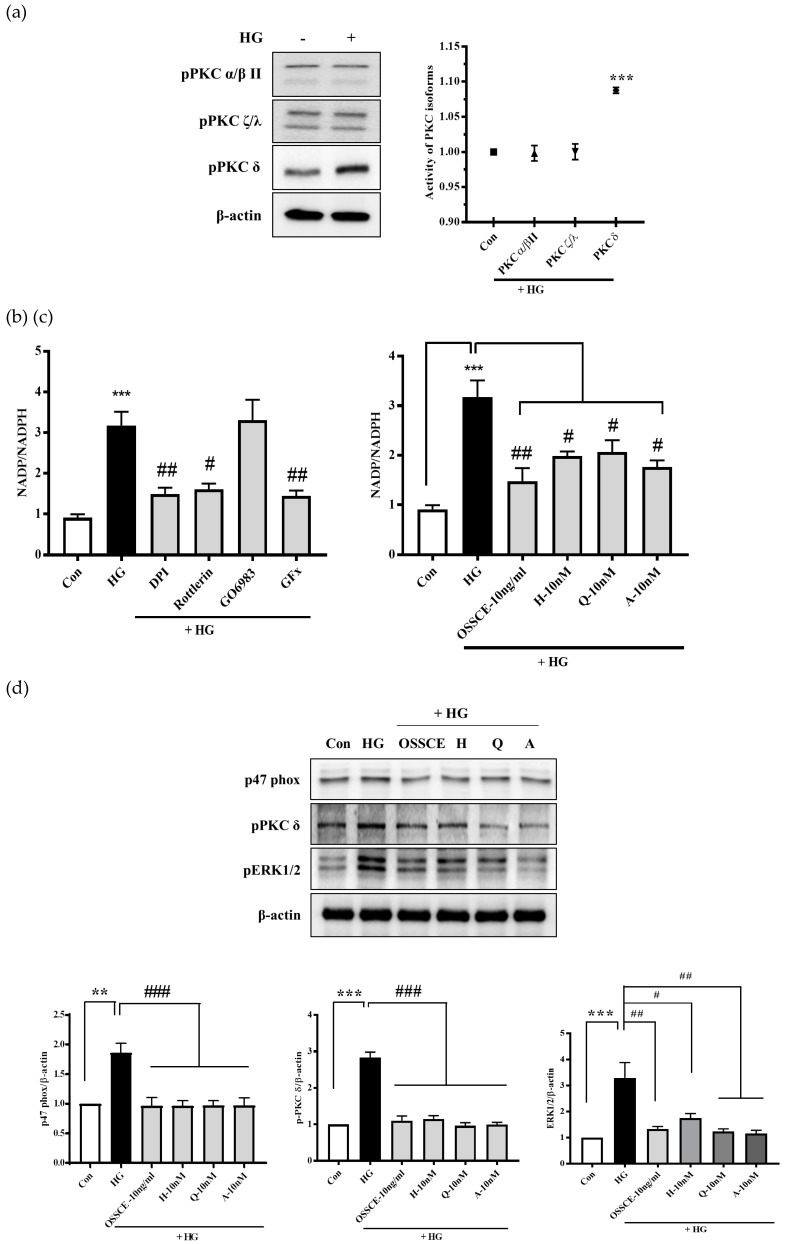
OSSCE and MCs inhibit the expression of HG-induced PKCδ, p47phox, and ERK1/2 in HRMECs. (**a**) The activity of PKC isoforms was evaluated. Only PKCδ was activated under HG-treated condition. (**b**) The activity of NADPH oxidase was inhibited by DPI (NADPH oxidase inhibitor), rottlerin (PKCδ inhibitor), and GFx (PKC inhibitor). G06983 (PKCα/βII inhibitor) did not affect the NADPH oxidase activity. (**c**) OSSCE and MCs inhibit the HG-induced NADPH oxidase activity in HRMECs. The activity of NADPH oxidase was measured by the luminescence assay. *** *p* < 0.001 vs. Con; ^#^
*p* < 0.05, ^##^
*p* < 0.01 vs. HG (*n* = 3). (**d**) The elevated expression of p47phox, ERK1/2, and PKCδ due to HG was significantly restored nearly to the normal range by the treatment with OSSCE and MCs. ** *p* < 0.01, *** *p* < 0.001 vs. Con; ^#^
*p* < 0.05, ^##^
*p* < 0.01, ^###^
*p* < 0.001 vs. HG. All the data are expressed as mean ± SD (*n* = 3).

**Table 1 nutrients-11-00546-t001:** The levels of HbA1c and blood glucose in SDT rats.

	Nor	SDT	OSSCE-100	OSSCE-250
Blood glucose (mg/dL)	144.1 ± 21.0	419.2 ± 21.1 *	393.4 ± 47.7	391.5 ± 52.2
HbA1c (%)	3.49 ± 0.07	9.13 ± 0.37 *	9.14 ± 0.30	8.74 ± 0.48

Nor, normal SD rat; SDT, Spontaneously diabetic Torii rat; OSSCE-100, SDT rat treated with 100 mg/kg OSSCE; OSSCE-250, SDT rat treated with 250 mg/mL OSSCE. All data were expressed as means ± SEM. * *p* < 0.01 vs. NOR group.

## Data Availability

All data generated or analysed during this study are included in this published article.

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
