# Peer review of "Improvement in Diabetic Retinopathy through Protection against Retinal Apoptosis in Spontaneously Diabetic Torii Rats Mediated by Ethanol Extract of Osteomeles schwerinae C.K. Schneid"

_nutrients, 2019, doi:10.3390/nu11030546_

Reviewer 1 Report

The manuscript presents data shedding light onto the mechanism of the protective role of OSSCE and in particular the effect of this traditional medicine on diabetic retinopathy in an animal model.  Overall it is an interesting paper and the experiments have, on the whole been prepared and presented accurately but there is room for improvement and suggestions to this effect follow:

The introduction could be shorter and sharper.

 There are a few errors:

- line 42 should be 'working age' rather than working class'.

- line 124 - reference is made to cells but details of cells used does not take place until line 162.

The methods section includes an outline of animal experimental design.  It would be beneficial to also include the design of the experiments with cells as the explanations in the results are difficult to understand.

Fig 2 a Comparison of OSSCE and aminoguanigine would be preferable on same graph using same axis scale.

Fig 2 d images of retina with corresponding image analysis of AGES.Legend information is insufficient to really understand what is going on in this very busy figure.

The same features are present in the remaining figures and it would be beneficial to address the same issues regarding consistency of presentation, clarity of images presented, legend details and generally improve the presentation of the data in a more logical, systematic way.

Discussion

Generally well written with minimal changes required.  There is a need to put in some attention to detail and a review of the text prior to submission is needed eg. line 460 no full stop.

Author Response

Comments and Suggestions for Authors

The manuscript presents data shedding light onto the mechanism of the protective role of OSSCE and in particular the effect of this traditional medicine on diabetic retinopathy in an animal model.  Overall it is an interesting paper and the experiments have, on the whole been prepared and presented accurately but there is room for improvement and suggestions to this effect follow:

-The introduction could be shorter and sharper. => “Introduction” was changed more shorter and clear.

 There are a few errors:

- line 42 should be 'working age' rather than 'working class'.=> Changed

- line 124 - reference is made to cells but details of cells used does not take place until line 162. .

=> Changed as following:

2.4. Inhibitory activity on AGE formation expression of RAGE in HRMECs

Human retinal microvascular endothelial cells (HRMECs) were purchased from Cell Systems (Cat. No. ACBRI 181, Kirkland, WA, USA) and used at passages 3–7. Cells were grown in CSC complete medium (CS-4ZO-500; Cell Systems) containing Bac-Off® (antibiotic). Cultures were maintained at 37°C in a humidified 95% air/5% CO2 atmosphere [24]. For the inhibitory activity on AGE formation and expression of RAGE, cells were treated with either OSSCE or AG dissolved in DMSO for 1 h before addition of 25 mM HG and 500 µg/mL BSA, following which they were incubated for 24 h. Cells were prepared for Western blot analysis.

2.4 Inhibitory activity on AGE formation and expression of RAGE in HRMECs

Cells were grown in DMEM/F-12 with 10% foetal bovine serum in a 5% CO2 incubator. They were treated with either OSSCE or AG dissolved in DMSO for 1 h before addition of 25 mM HG and 500 µg/mL BSA, following which they were incubated for 24 h. To obtain the protein, cells were lysed with Laemmli sample buffer (Bio-Rad) and heated at 95°C for 5 min. Protein was separated by SDS–PAGE and transferred to a PVDF membrane using a Bio-Rad semi-blotting apparatus. The membrane was incubated with antibodies specific for AGE (1:2000, Trans Genic Inc.), RAGE (1:5000, Cell signalling), and β-actin (1:3000, Cell signalling), washed, and incubated with horseradish peroxidase-linked secondary antibodies. All sample detection and analysis was performed using LAS-3000 (Fuji Photo Film).

2.6 Cell culture

Human retinal microvascular endothelial cells (HRMECs) were purchased from Cell Systems (Cat. No. ACBRI 181, Kirkland, WA, USA) and used at passages 3–7. Cells were grown in CSC complete medium (CS-4ZO-500; Cell Systems) containing Bac-Off® (antibiotic). Cultures were maintained at 37°C in a humidified 95% air/5% CO2 atmosphere [24]. For the inhibitory activity on AGE formation and expression of RAGE, cells were treated with either OSSCE or AG dissolved in DMSO for 1 h before addition of 25 mM HG and 500 µg/mL BSA, following which they were incubated for 24 h. Cells were prepared for Western blot analysis.

- The methods section includes an outline of animal experimental design.  It would be beneficial to also include the design of the experiments with cells as the explanations in the results are difficult to understand. => “Materials and Methods” section was changed easily to understand.

- Fig 2 a Comparison of OSSCE and aminoguanigine would be preferable on same graph using same axis scale. => IC50 values of OSSCE and aminoguandine (AG) are great difference (IC50 of OSSCE: 16.34 ± 0.04 µg/mL / IC50 of AG: 72.28 ± 4.21 µg/mL). It is not preferable to show the activities of both on same graph. However, the y axis scales of OSSCE and AG were up to 100%.

- Fig 2 d images of retina with corresponding image analysis of AGES. Legend information is insufficient to really understand what is going on in this very busy figure. => Fig2 legend was changed.

The same features are present in the remaining figures and it would be beneficial to address the same issues regarding consistency of presentation, clarity of images presented, legend details and generally improve the presentation of the data in a more logical, systematic way. => All were newly edited and improved.

Discussion

- Generally well written with minimal changes required.  There is a need to put in some attention to detail and a review of the text prior to submission is needed eg. line 460 no full stop. => Added the period after reference [30, 31].

Reviewer 2 Report

In this manuscript, the authors have examined Improvement in diabetic retinopathy through protection against retinal apoptosis in spontaneously diabetic Torii rats mediated by ethanol extract of Osteomeles schwerinae C.K. Schneid. The manuscript is very well written and experimental design is very good. Introduction, discussion are well written and study is well planned. However, after incorporating below mentioned changes manuscript can be accepted. Here, some points that authors should consider.

Major points:

1.    Figure 3 which depicts OSSCE inhibits retinal apoptosis in SDT rats, as well as ROS generation and 8-OHdG  expression in HG-treated HRMECs. Panel A does not make any sense to me. As it is well established that Bax and Bcl2 expression is opposite of each other. Authors are reporting the elevated expression of Bax and bcl2 in the same sample. It can not be possible and similarly, SDT have a more cleaved caspase-3 expression which it should have but it should corroborate with Bax expression. The author should explain this or provide some better blots.

2.    Similarly, figure 5 b has a problem. The densitometric analysis does not corroborate with representative blots. Cytochrome C expression does not seem to be increasing that prominently as presented in densitometric analysis. And in cytosol with Q+HG treatment, in fact, increasing the release of cytochrome c which should not be the case and OSSCE +HG should significantly decrease the cytochrome C expression. The decrease does not seem to be that evident. Please explain or provide some better blots. Which should clearly represent their storyline.

3.    Regarding gel shift assay studies, the author should describe EMSA in detail in the material method section. And also, if possible please provide the whole gel showing the free probe.

Minor points:

1.    Author should provide the reference for The retinal neuron cells begin to die soon after the onset of streptozotocin (STZ)-induced diabetes in an experimental rat model. The increase in frequency of apoptosis occurred after only one month of induction, and a similar increase was noted in human retinas after six years of diabetes.

2.    Please provide the source of Nf-kB antibody.

3.    Please incorporate some recent reference like Sharma I, Tupe RS, Wallner AK, Kanwar YS. Contribution of myo-inositol oxygenase in AGE:RAGE-mediated renal tubulointerstitial injury in the context of diabetic nephropathy. Am J Physiol Renal Physiol. 2017;314(1):F107-F121.

4.    Author should correct DAPI with TOPO in figure 6 b. DAPI is blue in color.

Author Response

Comments and Suggestions for Authors

In this manuscript, the authors have examined Improvement in diabetic retinopathy through protection against retinal apoptosis in spontaneously diabetic Torii rats mediated by ethanol extract of Osteomeles schwerinae C.K. Schneid. The manuscript is very well written and experimental design is very good. Introduction, discussion are well written and study is well planned. However, after incorporating below mentioned changes manuscript can be accepted. Here, some points that authors should consider.

Major points:

1.      Figure 3 which depicts OSSCE inhibits retinal apoptosis in SDT rats, as well as ROS generation and 8-OHdG expression in HG-treated HRMECs. Panel A does not make any sense to me. As it is well established that Bax and Bcl2 expression is opposite of each other. Authors are reporting the elevated expression of Bax and bcl2 in the same sample. It can not be possible and similarly, SDT have a more cleaved caspase-3 expression which it should have but it should corroborate with Bax expression. The author should explain this or provide some better blots. => Legend of Fig3 was changed and Panel b was relabeled. Bax image in Fig2c was changed

2.      Similarly, figure 5 b has a problem. The densitometric analysis does not corroborate with representative blots. Cytochrome C expression does not seem to be increasing that prominently as presented in densitometric analysis. And in cytosol with Q+HG treatment, in fact, increasing the release of cytochrome c which should not be the case and OSSCE +HG should significantly decrease the cytochrome C expression. The decrease does not seem to be that evident. Please explain or provide some better blots. Which should clearly represent their storyline. => As reviewer’s comments, we measured the band intensities again and images were analyzed with image J program. Bar graph was slightly changed and better blots images were changed (Fig. 5b).

3.    Regarding gel shift assay studies, the author should describe EMSA in detail in the material method section. And also, if possible please provide the whole gel showing the free probe. => provided in method section 2.14

Minor points:

1.    Author should provide the reference for The retinal neuron cells begin to die soon after the onset of streptozotocin (STZ)-induced diabetes in an experimental rat model. The increase in frequency of apoptosis occurred after only one month of induction, and a similar increase was noted in human retinas after six years of diabetes => ref.4 was cited for this.

2.    Please provide the source of Nf-kB antibody => provided in 2.14.

3.    Please incorporate some recent reference like Sharma I, Tupe RS, Wallner AK, Kanwar YS. Contribution of myo-inositol oxygenase in AGE:RAGE-mediated renal tubulointerstitial injury in the context of diabetic nephropathy. Am J Physiol Renal Physiol. 2017;314(1):F107-F121. => This paper (Sharma I, et al.,) was cited as ref. 10.

4.    Author should correct DAPI with TOPO in figure 6 b. DAPI is blue in color. => Originally, DAPI is blue color. Previous images showed red that we used DP manager program (from Olympus) to show clearly. Images in Fig 6b were changed blue.

Round  2

Reviewer 2 Report

Authors have incorporated the all changes as mentioned by the reviewers, However they should be more careful while preparing figures, please change Bax to Bcl2 in figure 3 (there can not be two Bax and showing reverse results).  Material and methods section is one of the most important part of a good manuscript. In future please try to explain as much you can. And for gel shift studies always provide full autoradiograph, equal amount of free probe should be there in each lane. Although Nf-kB is a very traditional transcription factor. Its implication is well established.  

Author Response

Bax in Figure 3c was changed into Bcl2.

Full autoradiograph of NF-kb gel shift in Figure 6a is provided.

In legend of Figure 2d, the meaning of GCL, INL and ONL are explained.